# Using Airborne Lidar, Multispectral Imagery, and Field Inventory Data to Estimate Basal Area, Volume, and Aboveground Biomass in Heterogeneous Mixed Species Forests: A Case Study in Southern Alabama

Schyler Brown [1,*], Lana L. Narine [1] and John Gilbert [2]

1   College of Forestry and Wildlife Sciences, Auburn University, 602 Duncan Dr., Auburn, AL 36849, USA; lln0005@auburn.edu
2   Solon Dixon Forestry Education Center, Auburn University, 12130 Dixon Road Center, Andalusia, AL 36420, USA; gilbejo@auburn.edu
*   Correspondence: sbb0056@auburn.edu

**Abstract:** Airborne light detection and ranging (lidar) has proven to be a useful data source for estimating forest inventory metrics such as basal area (BA), volume, and aboveground biomass (AGB) and for producing wall-to-wall maps for validation of satellite-derived estimates of forest measures. However, some studies have shown that in mixed forests, estimates of forest inventory derived from lidar can be less accurate due to the high variability of growth patterns in multispecies forests. The goal of this study is to produce more accurate wall-to-wall reference maps in mixed forest stands by introducing variables from multispectral imagery into lidar models. Both parametric (multiple linear regression) and non-parametric (Random Forests) modeling techniques were used to estimate BA, volume, and AGB in mixed-species forests in Southern Alabama. Models from Random Forests and linear regression were competitive with one another; neither approach produced substantially better models. Of the best models produced from linear regression, all included a variable for multispectral imagery, though models with only lidar variables were nearly as sufficient for estimating BA, volume, and AGB. In Random Forests modeling, the most important variables were those derived from lidar. The following accuracy was achieved for linear regression model estimates: BA $R^2$ = 0.36, %RMSE = 31.26, volume $R^2$ = 0.45, %RMSE = 35.30, and AGB $R^2$ = 0.41, %RMSE = 31.31. The results of this study show that the addition of multispectral imagery is not substantially beneficial for improving estimates of BA, volume, and AGB in mixed forests and suggests that the investigation of other variables to explain forest variability is necessary.

**Keywords:** laser; forestry; allometry; parametric; non-parametric; modeling 3DEP; Alabama

## 1. Introduction

Estimates of forest inventory such as basal area (BA), volume, and aboveground biomass (AGB) across large tracts of land provide foresters and ecologists the information necessary to form and implement management strategies at both small and large scales [1,2]. For example, BA (a measure of the cross-sectional area of a tree at breast height) has traditionally been used to manage naturally regenerated forest stands for timber production. BA is also linked to other forest metrics and can be used to estimate volume and biomass [3–6]. BA can also be a useful forest measurement for ecological studies, and a perfect example to demonstrate this point is found in longleaf pine (*Pinus palustris*) forests in the U.S. Southeast. In the U.S. Southeastern, it is not only necessary to know the amount of wooded area of the longleaf pine for conservation of the species, but Red-Cockaded woodpeckers (*Leuconotopicus borealis*) depend on a balance of longleaf pine basal area and stand density that are suitable for cavity nesting [7]. BA is also useful for examining the

amount of woody area infested by various species of invasive insects across large areas [8]. Volume is a measure used by foresters to estimate the cubic amount of wood on an area of land [9,10]. An estimate of volume can give foresters an estimate of the dollar value of the standing timber they are cruising. Volume is also associated with the persistence of invasive species, and having estimates of volume can help foresters better manage both timber product and ecosystem health through stem and growth stocking projections [11]. AGB is the dry weight of carbon stored within forest trees above the ground and is measured in Mg ha$^{-1}$ [12]. In ecological studies, AGB is a known driver of the species composition of an ecosystem, and maps of AGB can give ecologists insights into the distribution and composition of organisms across great extents [13]. Because AGB is an estimation of the carbon stored in the trunks of trees, it is critical to understand the role forests play in carbon cycling and climate change. Accurate estimates of AGB can help nations develop strategies to meet goals set by international agreements for climate change, such as those outlined in the Reducing Emissions from Deforestation section of the Paris Climate Agreement [14], and contributing action inventories such as the National Greenhouse Gas Inventories report [15]. Often it can be difficult or impossible to accurately estimate AGB without destructively sampling trees in a forest, and allometric equations developed from field inventory estimating BA, volume, and AGB can fail to account for variables that affect them across a large landscape, such as site index and crowding, among other local factors [2,16]. Inconsistencies in forest inventory methods across the U.S. also cause inaccuracies in regional estimates, as inventory methods are frequently tailored for project-specific goals and thus have differing methods for measuring [17–19]. Lastly, estimates of forest measurements across large areas using traditional mensuration approaches can be time-consuming and costly, and in some instances, variation in landscape features can cause dangers in field work and inaccuracies in estimates [16,18,20–22]. Remote sensing (RS), particularly light detection and ranging (lidar) estimates of BA, volume, and AGB, may overcome these limitations and can produce comparable [23] and more accurate results [24] compared to field-based estimates of large-area forest inventory.

Lidar, an RS method that uses laser scanning to acquire three-dimensional information over the desired area [25], provides an alternative approach to estimating forest inventory metrics [1,21]. By using lidar, forest characteristics such as tree heights and canopy cover can be directly estimated, and other measures such as BA, volume, and AGB can be estimated by means of modeling using a variety of metrics derived from directly measured height and canopy cover [26,27]. The most commonly used approach to modeling BA, volume, and AGB from lidar data is by developing a multiple linear regression from field inventory and lidar variables [28]. Often the best subsets approach, such as forward or backward selection, is used to determine a model with variables that best explain the dependent variable and further validated using a cross-validation or set validation approach [27,29,30]. These models usually contain at least one variable explaining tree height, another explaining canopy cover, and another that accounts for variation in the data, such as a height standard deviation variable [6,31,32]; however, the variables in these models may differ depending on the study site and the foliage type being measured [28,33,34]. Estimates of BA, volume, and AGB can also be acquired from non-parametric machine learning approaches such as random forest (RF), a machine learning algorithm that uses random and iterative samples of the data to produce regression trees and bootstraps data for robust predictive models [35–37].

Estimating forest metrics may be performed by using an area-based approach or a tree-level approach, and previous studies modeling BA, volume, or AGB have seen success in both approaches. For example, in estimating forest attributes from individual trees, researchers found that BA, volume, and tree heights of longleaf pine in Georgia, U.S., could be estimated with accuracies of $R^2$ = 0.18, 0.94, and 0.96, respectively [38]. The results for estimating volume and tree heights of individual trees are promising, and the poor results for estimating BA were explained by the loss of height-diameter allometry in southern pines above 25 m. In another study, researchers estimated BA for a plantation of loblolly pines (*Pinus taeda*) (another southern pine species) using an area-based approach

and were able to achieve accuracies of $R^2 = 0.97$, and noted that the highly homogeneous environment of the pine plantations that the trees were grown in likely led to such high accuracies [27]. Using an area-based approach across a large area can be beneficial in that it reduces the amount of processing required by the computer. Similarly, variables can be derived from multispectral imagery, such as the normalized difference vegetation index (NDVI) and texture co-occurrence, which have also proven useful for modeling [34]. While predictive modeling using variables from either lidar or multispectral imagery has been successful in many cases, particularly in northern and western forests where pine species dominate [17,30], more difficulty is found in estimating forest inventories in heterogeneously mixed forests [28,34,38]. This, in addition to the perceived cost of remote sensing data, has caused delays in adopting RS technology for use in some forest inventories. However, the combination of lidar data, multispectral imagery, and field data for modeling could improve estimates of BA, volume, and AGB in mixed forests [39–41]. Furthermore, open-source data are becoming more available and can be used for forest inventories in place of otherwise expensive data sources. Finally, the implications of achieving accurate estimates of forest inventory metrics go beyond timber cruising and carbon estimation alone. By achieving wall-to-wall estimates of forest inventory metrics, past and future data can be compared for forest management techniques and landscape ecology analysis. Most importantly, the wall-to-wall maps produced in this study can be used as base maps for validation of satellite lidar data produced by systems such as the Global Ecosystem Dynamics Investigation (GEDI) lidar aboard the International Space Station (ISS) [42]. The goal of this study was to produce wall-to-wall estimates of BA, volume, and AGB in a southern mixed-species forest using open-source RS data. This goal was achieved by meeting the following objectives:

A. Determine whether the addition of variables derived from multispectral imagery to models previously including only lidar derived variables can improve estimates of BA, volume, and AGB;

B. Determine what variables are useful for modeling each forest metric (BA, volume, AGB) for Southern mixed forests.

If sufficiently accurate, models predicting forest inventory metrics can be used a number of times in forest inventory [17], and final wall-to-wall map outputs can be used for the validation of satellite data.

## 2. Materials and Methods

### 2.1. Study Area

The study area is a 35.35 km$^2$ site in Covington and Escambia counties, Southern Alabama (Figure 1). The site is within one of seven-level III ecoregions in Alabama, known as the Southern Plains ecoregion (ecoregion 65), whose natural vegetation is described as southern mixed pine forests and whose geography includes rolling hills and a warmer temperate climate [43]. The region can be further described at the level IV ecoregion (65f) as having loamy acidic soils and dark tea-colored streams with mixed forests interspersed with pine plantations [43–45]. Southern pine species that persist and typically dominate in this region include Longleaf pine (*Pinus palustris*), Shortleaf pine (*Pinus echinate*), Slash pine (*Pinus elliottii*), Loblolly pine (*Pinus taeda*), and Virginia pine (*Pinus virginiana*). The mean elevation of the study site is 59.47 m, with a minimum and maximum elevation ranging from 25 to 99 m, respectively, and it receives approximately 135 cm of rainfall a year. Within the study area is the Solon Dixon Forestry Education Center (SDFEC), a multi-use facility (outreach, extension, teaching, and research) primarily focused on providing hands-on field training to university students. Forest inventory data for this study come from the SDFEC and are described in more detail in the following sub-sections.

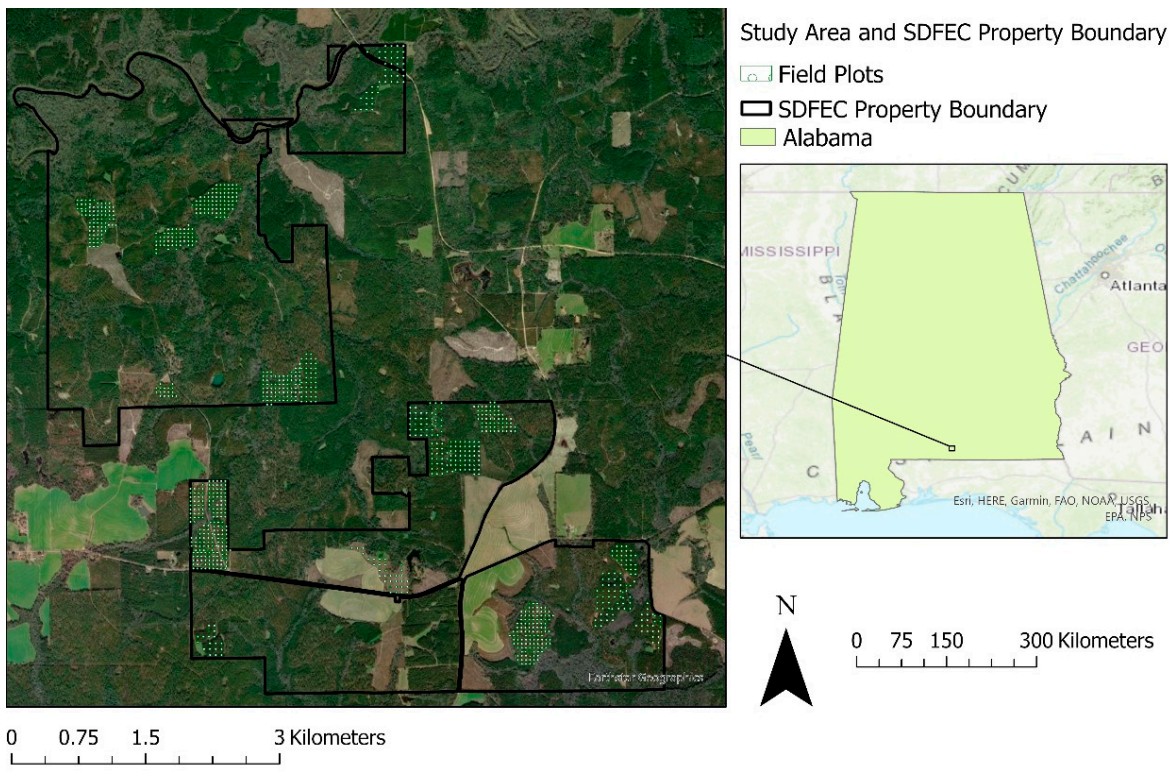

**Figure 1.** Study site with SDFEC property boundary and field plots, ESRI "World Imagery" [46].

### 2.1.1. Inventory Data

The SDFEC consists of both naturally regenerated and planted pine, hardwood, and mixed pine-hardwood forest stands intended for timber production. Forest inventory data from the SDFEC were collected from 1048 circular plots of 0.04047 hectares (1/10th acre) size in 304 forest stands (see Figure 1). The number of plots used in this study was reduced from 1048 to 523 to include only plots that were inventoried within two years of the 2017 lidar and imagery data acquisition. Plots are generated using TCruise software [47] at approximately one plot per 0.4047 hectares (one plot per acre) with a distance between plot centers of approximately 44.20 to 51.82 m (145.0 to 170.0 ft). The data included latitude and longitude of plot centers, plot area (0.04047-hectare plots), merchantable timber heights of individual trees (meters), diameter at breast height for individual trees (centimeters), and plot BA ($m^2\ ha^{-1}$). The plot volume ($m^3\ ha^{-1}$) and plot sum of AGB ($Mg\ ha^{-1}$) were later calculated (see processing approach). The SDFEC uses lumber mill specifications when taking plot data on a forest inventory cruise. Trees surveyed in a plot must meet a minimum diameter at breast height (DBH) of 11.68 cm (4.598 inches) for a 12.70 cm (5.0 inches) class while being at least 4.600 m in height to a 7.620 cm (3.000 inches) top. Trees within plots that do not meet these specifications are not tallied. Furthermore, the SDFEC only tallies trees that local forest product mills are buying, so a variety of species, including Eastern Red Cedar (*Juniperus virginiana*) and Southern Black Cherry (*Prunus serotina*), are not counted.

### 2.1.2. Airborne Lidar

Airborne lidar data used in this study come from the United States Geological Survey (USGS) 3D Elevation Program (3DEP) [48]. The 3DEP is an open-source program originally designed to collect three-dimensional geographic elevation data for a given landscape; however, the data have been used in many fields, including forestry and ecology. The data used in this study were collected by a Leica ALS70-HP RS device mounted in a fixed-wing position aboard Cessna and Partenavia aircrafts. Characteristics of the flight included a scan angle of 45 degrees, a flight height of 2000 AGL meters, and a speed of 130 kts. Lidar scanning characteristics included a 1 m footprint, a pulse density of 2.4 $pls/m^2$, and a scan

frequency of 35.1 Hz. The scan pattern was triangular. Fifty-four lidar files containing point cloud data collected in 2017 were downloaded from 3DEP using UGet software [49]. Each file contained a las and/or laz tile with associated metadata.

### 2.1.3. Multispectral Imagery

Multispectral imagery data were acquired from Copernicus Sentinel-2. The Copernicus Sentinel-2 mission was developed by the European Space Agency (ESA) with the purpose of providing global and continuous land cover data in addition to map products such as change detection products and land cover maps. The mission aims to provide data for studies focused on climate change, land management, and security [50]. Leaf-on and leaf-off Sentinel-2A Level-1C (L1C) top-of-atmosphere (TOA) reflectance orthoimages were downloaded from USGS EarthExplorer for the dates 19.VII.2016 and 24.II.2017. L1C products are derived from the L1B product and use radiometric and geometric corrections to produce the final product. The downloaded Sentinel-2 imagery consists of a ten-day, ten-meter spatial resolution for four spectral bands, including blue band 2 (490 nm), green band 3 (560 nm), red band 4 (665 nm), and NIR band 8 (842 nm). Imagery for this study consisted of less than twenty percent cloud cover.

### 2.2. Data Processing Approach

Basal area, volume, and AGB were calculated from field inventory data. Variables were derived from lidar and multispectral imagery for the study site. These variables were also extracted from within the boundaries of the field inventory plots for further modeling. A workflow is displayed in Figure 2.

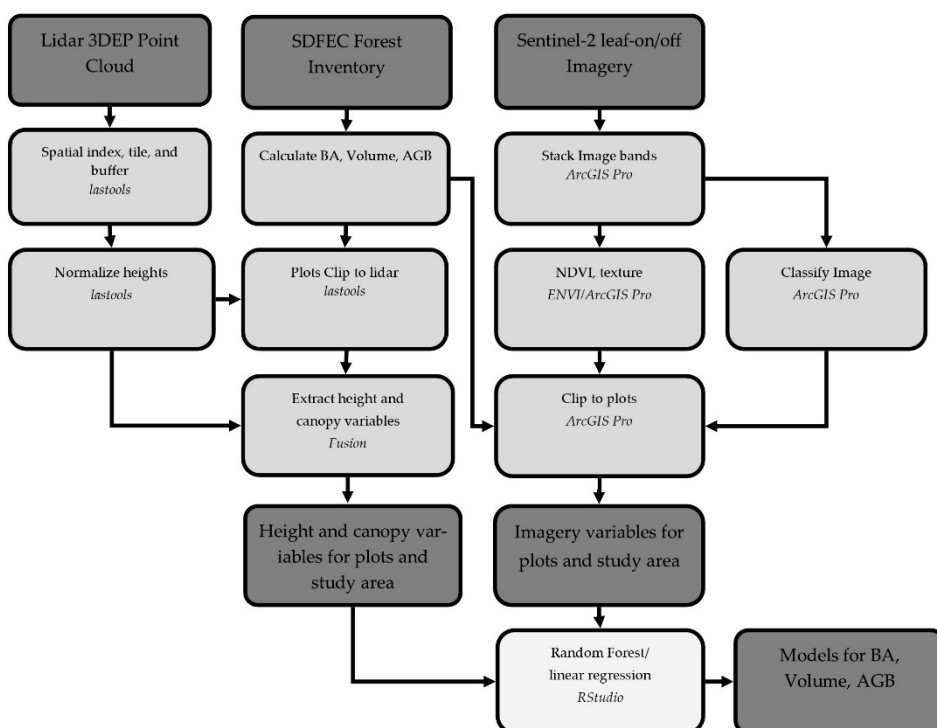

**Figure 2.** Data processing workflow.

### 2.2.1. Field Inventory

Equations for calculating plot BA, volume, and AGB are shown in Equations (1)–(3), respectively [4,10,12]. Summary statistics of the forest inventory are displayed in Table 1.

**Table 1.** Summary statistics of inventory plots.

|  | BA (m$^2$ ha$^{-1}$) | Volume (m$^3$ ha$^{-1}$) | AGB (Mg ha$^{-1}$) |
|---|---|---|---|
| Min | 1.583 | 4.253 | 5.498 |
| Max | 45.92 | 412.8 | 226.4 |
| Mean | 18.33 | 193.7 | 80.84 |
| Std. dev. | 7.121 | 68.53 | 33.27 |

Basal area was calculated using the following equation

$$\text{Sum of BA}_n = \sum((0.00007854 \times \text{DBH}^2)/0.04)_i \tag{1}$$

where DBH is equal to the diameter (centimeters) at breast height (1.30 m), 0.00007854 is a forester's constant (calculated from π/(40,000)) that converts centimeters to meters squared, and "i" is a tree within plot n [4]. The summed BA per plot was then calculated on a per hectare basis by dividing the value by the plot size (0.04 ha).

In order to match specifications from the DBH and height data from the SDFEC, a merchantable volume equation (Equation (2)) was used to calculate volume on a per hectare basis for hardwoods and pines

$$\text{Sum of Volume (m}^3 \text{ ha}^{-1}) = \sum(\alpha + \beta(\text{DBH}^2 \times \text{Ht}))_i \tag{2}$$

where α and β are species-specific parameters estimated from the equation [10]. In this study, species were broadly grouped into pine and hardwood groups. For pines, the longleaf pine parameters were used because they constitute the majority of pine BA, and for hardwood species, the "unknown hardwood" parameters were used because species-specific information for hardwoods was unavailable. The parameters for each group are displayed in Table 2. Lastly, DBH is equal to the diameter at breast height (centimeters), and Ht is the merchantable height of the tree.

**Table 2.** Species-specific parameters for calculating volume.

|  | Pines | Hardwoods |
|---|---|---|
| α | −0.4432 | 0.8235 |
| β | 0.002165 | 0.001630 |

Finally, aboveground biomass was calculated on a per hectare basis using the following

$$\text{Sum of AGB (Mg ha}^{-1}) = \sum(\text{Exp}(\beta_0 + \beta_1 \ln \text{DBH})) \tag{3}$$

where β0 and β1 are species specific parameters (Table 3) [12].

**Table 3.** Species-specific parameters for calculating AGB.

|  | Pines | Hardwoods |
|---|---|---|
| $\beta_0$ | −2.536 | −2.480 |
| $\beta_1$ | 2.435 | 2.483 |

2.2.2. Lidar

Lidar data were processed using LASTools [51] and FUSION [52] software. In LAS-Tools, aboveground tree heights were computed by normalizing points classified as ground returns and measuring the vegetation above ground points. Point cloud data were then clipped to the extent of field inventory plots in ArcGIS Pro. In FUSION, height and canopy

metrics were calculated based on the normalized point cloud and the clipped plots. Plot field inventory data were combined with lidar metrics for modeling. The slope was calculated from a DEM derived from the point cloud and extracted per plot in ArcGIS Pro [53].

### 2.2.3. Multispectral Imagery Processing

Sentinel-2 imagery data were used to derive multispectral variables of interest, including normalized difference vegetation index (NDVI) and second-order co-occurrence texture (Table 4). NDVI is known as a greenness vegetation index ratio with a range of −1 to 1 [54]. NDVI was calculated in ENVI [54] using the following equation

$$NDVI = (NIR - Red)/(NIR + Red) \tag{4}$$

where NIR and red are the near-infrared and red band layers derived from the imagery. Second-order co-occurrence texture shows the relationship between one pixel value and its surrounding neighbors based on distance and angularity [55]. These metrics and NDVI were derived in ENVI as 20 m pixel resolution rasters. The means of these values were further extracted from the zone of each field plot in ArcGIS Pro using the zonal statistics as a table tool in ArcGIS Pro.

**Table 4.** Variables used in modeling. Note that repetitive variables such as height percentiles and texture metrics for each of 4 bands were grouped to reduce table size.

| Variable | Description |
| --- | --- |
| 1. Max | Maximum height for a tree in a given plot |
| 2. Mean | Mean height of trees in a given plot |
| 3. Variance | Variance of tree heights in a plot |
| 4. ElevCV | Co-variance of heights in a plot |
| 5. p10, 25, 50, 75, 90, 95, 99 | Height percentiles of trees in a plot |
| 6. cancovA | % first returns above 4.6 m |
| 7. cancovB | % all returns above 4.6 m |
| 8. dens_0_10 | Count of returns within 0–10 m |
| 9. dens_10_15 | Count of returns within 10–15 m |
| 10. dens_15_20 | Count of returns within 15–20 m |
| 11. stddev | Standard deviation of tree heights |
| 12. con_1,2,3,4 | Texture contrast for bands r,b,g,NIR |
| 13. cor _1,2,3,4 | Texture correlation for bands r,b,g,NIR |
| 14. dis _1,2,3,4 | Texture dissimilarity for bands r,b,g,NIR |
| 15. ent _1,2,3,4 | Texture entropy for bands r,b,g,NIR |
| 16. hom _1,2,3,4 | Texture homogeneity for bands r,b,g,NIR |
| 17. mean_1,2,3,4 | Mean texture for bands r,b,g,NIR |
| 18. secmom_1,2,3,4 | Angular second moment for bands r,b,g,NIR |
| 19. var_1,2,3,4 | Texture variance for bands r,b,g,NIR |
| 20. NDVI | NDVI within a plot |
| 21. Land_cover | Landcover classes including Deciduous, Evergreen, and Barren |
| 22. Slope | Slope calculated as a degree of inclination |

With a temporal resolution of 10 days, Sentinel-2 data have the advantage of leaf-on and leaf-off acquisitions and made it possible to develop a landcover map distinguishing coniferous and deciduous vegetation in the SDFEC. This was achieved by stacking four bands (blue, green, red, and NIR) from each season (leaf-on and -off) and a canopy height model (CHM) derived from the lidar together, resulting in a nine-band image. Regions of interest were gathered from bands representing leaf-on and leaf-off years, and by using maximum likelihood supervised classification, a landcover map was produced with the following cover types: water, barren land, agriculture, deciduous vegetation, and evergreen vegetation. A confusion matrix was used to assess the accuracy of the supervised classification. The kappa coefficient, an indicator of how the classification of an image compares to ground truth data, was computed from the confusion matrix. This value ranges from 0 to 1, where 0 suggests that no none of the pixels were correctly classified, and 1 indicates that

all pixels were correctly classified. The kappa for the classified image was 0.74, and the user accuracies were highest in the water and evergreen classes (user accuracy > 0.95), with the most confusion occurring in the agricultural and barren land cover classes. The highest producer accuracy occurred in evergreen and deciduous classes (0.73 and 1.00, respectively). Land cover types were extracted from each forest inventory field plot. A plot was then assigned a land cover type based on the dominant land cover type in that plot. Table 4 lists the 52 lidar, spectral, textural, and land cover metrics used in this study.

### 2.3. Data Analysis

Two modeling approaches were applied for predicting BA, volume, and AGB: multiple linear regression and Random Forest (RF). Both modeling approaches were implemented using the R programming language in RStudio [56]. Before modeling via linear regression or RF, the categorical landcover variables needed to be dummy coded to be used in linear regression modeling. Therefore, two data sets were created for each dependent variable (BA, volume, and AGB), one with dummy coded land cover variables and one with categorical land cover variables.

### 2.3.1. Regression Modeling

For the multiple linear regression approach, the regsubsets function from the leaps package in R [57] was used to select variables and develop models for BA, volume, and AGB using forward and backward selection. In forward variable selection, a model is produced by starting with one variable and adding variables to the model until the best possible model is produced. Conversely, the backward variable selection includes every variable in the model and removes a variable until the best model is selected. The best five models were developed for up to the best five variable models based on the highest $R^2$, lowest Mallow's cp, and lowest Bayesian Information Criterion (BIC). A Breusch–Pagan test determined that the response variables were not normally distributed, so a log transformation was applied to each [58]. In order to avoid multicollinearity, variance inflation factors (VIF) were calculated for each variable in each model, and models whose predictor variable's VIFs exceeded 10 were be removed [59]. A threshold of 0.05 was used to determine variable significance.

### 2.3.2. Random Forest (RF)

RF is a non-parametric machine learning modeling technique that uses random bagging and a bootstrap sample of the data to create a number of user-specified trees that vote on the best model parameters at each node of each tree [35]. The major benefit of RF in this study is that it can handle a large number of variables and work with data that may otherwise push the boundaries of the assumptions made by multiple linear regression, such as the assumption that there is no multicollinearity. The "RF" function in the ModelMap, a package in R [60], was used for modeling BA, volume, and AGB. This package easily allows users to input data and tune hyperparameters to obtain a specified raster output. Important user set hyperparameters for RF include the mtry and ntree parameters. The mtry parameter is the number of randomly selected variables at each node and was automatically optimized. The ntree parameter is the number of trees grown in the model and was set to 500 trees. After model building, the model can be used to predict BA, volume, and AGB across the study extent. Raster layers for each lidar and imagery independent variable were created for the extent of the study site with the same number of rows and columns at a 20 m resolution. These rasters were referenced using a look-up table (LUT), and the values for each cell are entered into the RF model to produce an estimation of the desired metric across the study site.

### 2.3.3. Model Evaluation

Selected linear regression models were evaluated using 10-fold cross-validation. This approach first randomly divides the dataset into 10 groups (folds) of equal sizes and then

tests the data on one group using 1–10 folds to fit the data. The process repeats k number of times, computing the mean squared error (MSE) for each test group and averages the MSE of all tests to obtain the k-fold cross-validation estimate [61]. While leaving one out, cross-validation is a common model evaluation approach [27,62,63]; it was shown that k-fold cross-validation could reduce the test error rate [61]. After k-fold validation, the predicted values are back-transformed and compared to the observed values. The final model used for mapping was selected based on the $R^2$, RMSE, percent root mean squared error (%RMSE), mean absolute error (MAE), and bias. The resulting models were used to create wall-to-wall maps of the independent variables (BA, volume, AGB) using the raster package in R [64]. F-tests were used to determine if models with imagery-derived variables significantly improved the model, and $R^2$ and %RMSE were compared.

In evaluating the RF model, the out-of-bag error (OOB) was used to determine the prediction error. In the RF model building process, random bootstrap samples were used to build decision trees. Some of the data were left out of each sample; this is the OOB sample. The OOB error is the amount of error produced from wrongly predicted OOB samples. Evaluation of the RF models included calculating the OOB RMSE, %RMSE, $R^2$, MAE, and bias. In order to evaluate whether imagery-derived variables were useful in the RF model, variable importance plots were produced. Importance is calculated as the decrease in model accuracy as variables are removed, or the percent increase in mean square error (%IncMSE).

### 3. Results

#### 3.1. Linear Regression Model

From the stepwise variable selection, the forward selection approach produced the best models, followed by backward selection. A table of the five best models for each response variable is displayed in Appendix A (Table A1). According to the calculated vif's, no variable in a model contributed to multicollinearity (VIF < 5). *p*-values indicated that each variable was significant to each of the models (*p* < 0.05). Models produced for BA, volume, and AGB in this study generally included a lidar height, canopy, and density metric, though BA did not include any lidar-derived height metric. Beyond three-variable models, the var_3 variable was included in the four- and five-variable model for BA. In the four- and five-variable model for volume, the hom_1 variable was included in addition to another lidar-derived density metric. Four and five variable models for AGB included var_2 and hom_1 variables, respectively, and the five-variable model included both.

The final model for volume outperformed those of BA and AGB. In model building, the five-variable model for volume had the lowest Mallow's cp and BIC and the highest $R^2$. After evaluating models from the 10-fold cross-validation approach, the five-variable model had the lowest error (RMSE = 49.23 m$^3$ ha$^{-1}$, %RMSE = 35.30) and the highest accuracy between the observed and predicted values ($R^2$= 0.45). This model included the following variables: p50, cancovA, dens_15_20, ElevCV, and hom_1. An F-test revealed that the hom_1 variable in the best model significantly improved the model (*p* < 0.05); however, the $R^2$ and %RMSE were not substantially different; the model without the 1_hom variable had the following accuracy: $R^2$ = 0.44; %RMSE = 35.96.

For BA, the model with the lowest BIC, Mallow's cp, and highest $R^2$ was the model with five variables. After validating BA using the 10-fold cross-validation approach, the model with five variables had the least amount of error (RMSE = 5.731 m$^2$ ha$^{-1}$, %RMSE = 31.26). Furthermore, the predictive accuracy between predicted and observed values using the five-variable model was highest ($R^2$ = 0.36). This model included the following variables: cancovA, dens_0_10, dens_10_15, dens_15_20, and var_3. An F-test revealed that the var_3 variable in the best model significantly improved the model (*p* < 0.05); however, the $R^2$ and %RMSE were not substantially different, and the model without the var_3 variable had the following accuracy: $R^2$ = 0.35; %RMSE = 31.61.

In modeling AGB, the model with the lowest BIC, Mallow's cp, and highest $R^2$ was the five-variable model. The five-variable model from the 10-fold cross-validation approach

also had the lowest RMSE (25.20 Mg ha$^{-1}$, %RMSE = 31.26%) and the highest correlation between observed and predicted values ($R^2$ = 0.41). The following variables were included in this model: p50, cancovA, dens_15_20, var_2, and hom_1. An F-test was used to determine if the addition of one or both of the image-derived variables significantly improved the model. The addition of one or both of the image-derived variables was significant according to the F-test ($p < 0.05$); however, the $R^2$ and %RMSE for all three models were not substantially different, the model without hom_1 had an accuracy of $R^2$ =0.41; %RMSE = 31.60, the model without both hom_1 and var_2 had $R^2$ =0.38; %RMSE = 32.09. Selected models with coefficients are presented in Table 5, and scatterplots showing model prediction and observed values are displayed in Figure 3 with map outputs at a 20 m resolution (approximately the size of the field inventory plots).

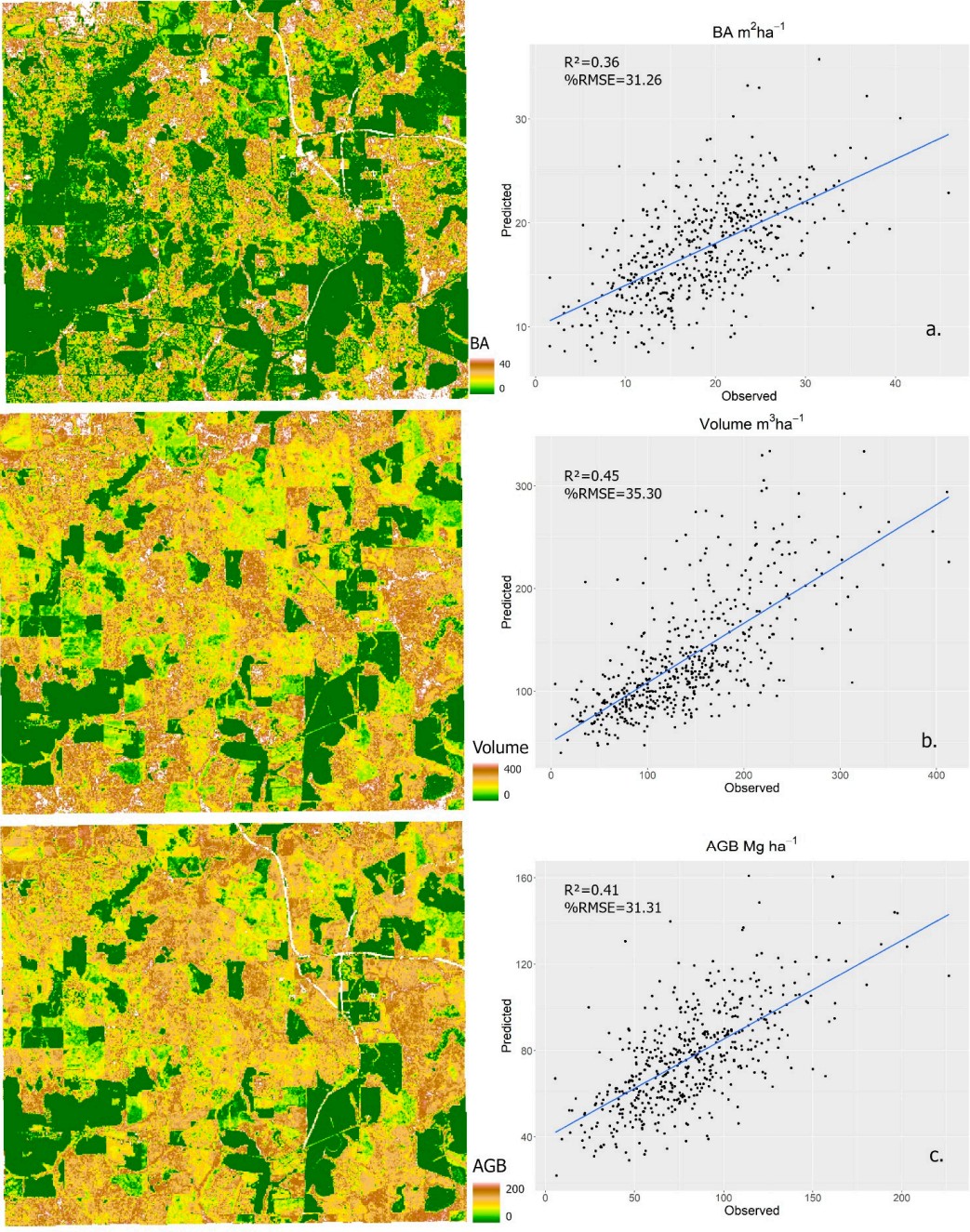

**Figure 3.** Predicted map outputs of BA (**a**), volume (**b**), and AGB (**c**) with scatterplot displaying correlation between observed and predicted values for the selected linear regression model.

**Table 5.** Models predicting BA, Volume, and AGB.

| Forest Metric | Model | %RMSE | RMSE | MAE | $R^2$ | Bias |
|---|---|---|---|---|---|---|
| BA | BA = 1.78 + 0.0173cancovA − 1.73dens_0_10 − 1.06dens_10_15 + 0.672dens_15_20 + 0.0245var_3 | 31.26 | 5.731 | 4.456 | 0.36 | 0.9528 |
| Volume | Vol = 3.09 + 0.0769p50 + 0.0145cancovA - 1.03dens_10_15 − 0.590hom_1 + 0.213Elev_CV | 35.30 | 49.23 | 37.32 | 0.45 | 8.287 |
| AGB | AGB = 2.53 + 0.0609p50 + 0.0136cancovA + 0.924dens_15_20 + 0.0406var_2 − 0.432hom_1 | 31.26 | 25.20 | 19.35 | 0.41 | 4.064 |

*3.2. Random Forest Model*

The RF models produced accuracies similar to those from the linear regression models. Model diagnostics revealed that volume again had the highest model accuracy ($R^2$ = 0.53, RMSE = 50.21 m$^3$ ha$^{-1}$ %RMSE = 36.68), followed by BA ($R^2$ = 0.39, RMSE = 5.662 m$^2$ ha$^{-1}$ %RMSE = 30.17) and AGB ($R^2$ = 0.37, RMSE = 26.29 Mg ha$^{-1}$ %RMSE = 34.26). Table 6 includes model evaluation from the observed vs. predicted values. Variable importance was calculated and plotted (Figure 4). Figure 5 display scatterplots for observed and predicted BA, volume, and AGB from random forest modeling.

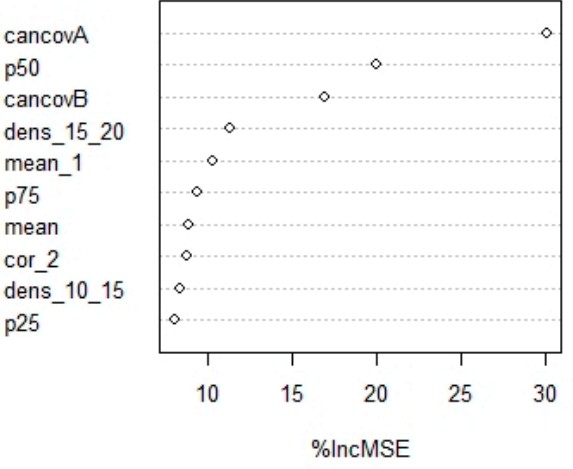

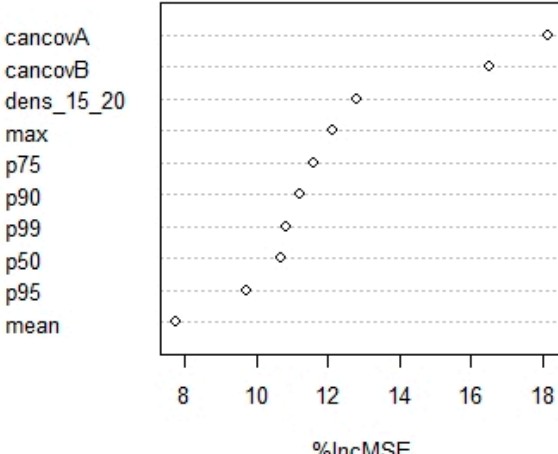

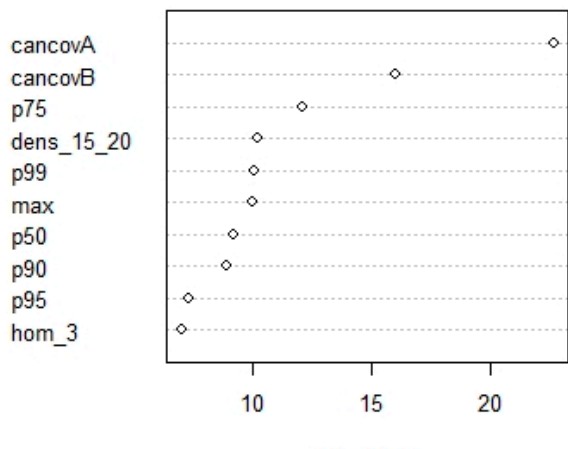

**Figure 4.** Importance plots for variables used in RF models.

**Table 6.** Model evaluation of the RF predictions.

| RF Model | %RMSE | RMSE | MAE | $R^2$ | Bias |
|---|---|---|---|---|---|
| BA | 30.17 | 5.662 | 4.449 | 0.39 | 0.01053 |
| Volume | 36.68 | 50.21 | 38.18 | 0.53 | −6.438 |
| AGB | 34.26 | 26.29 | 20.77 | 0.37 | −4.656 |

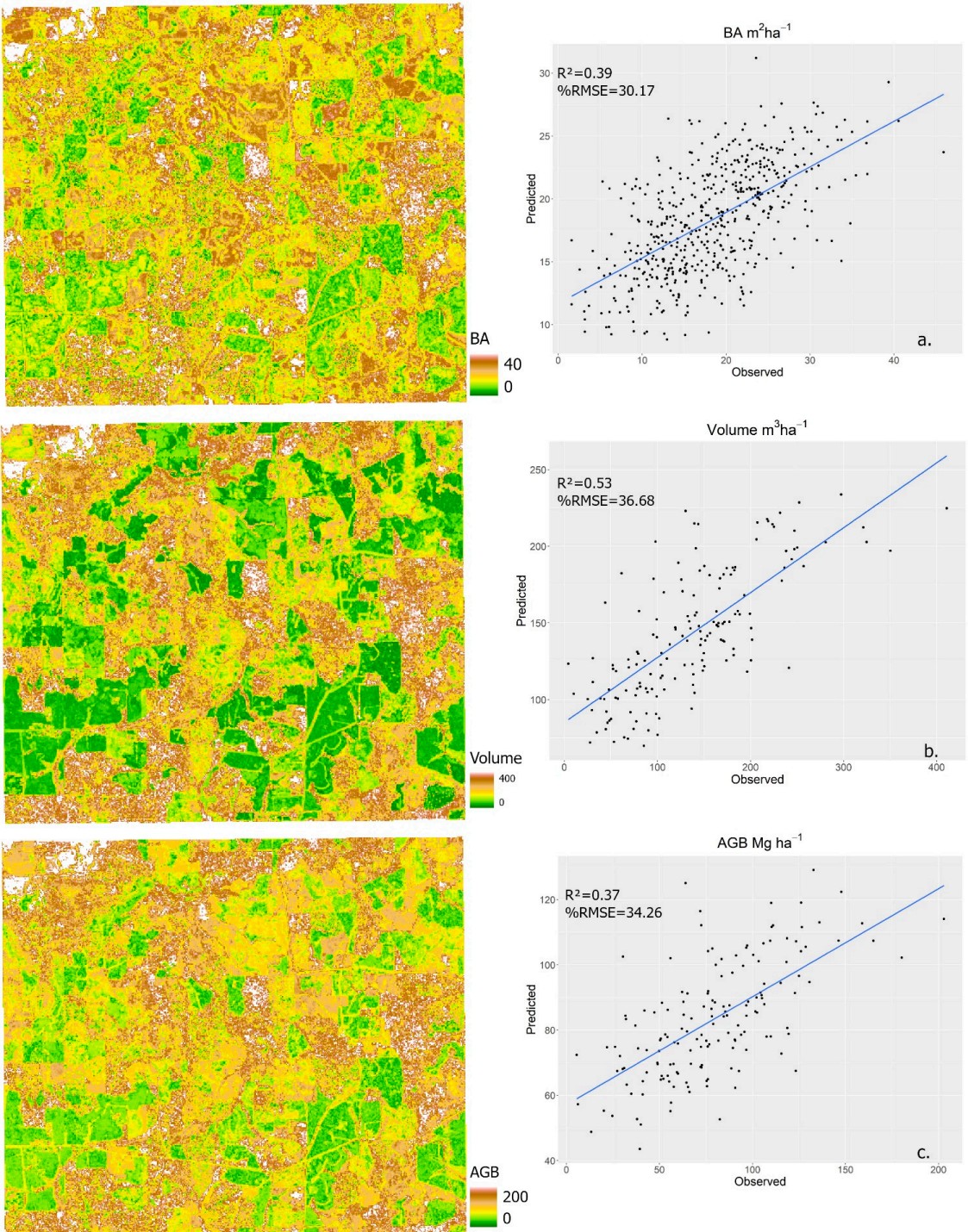

**Figure 5.** Predicted map outputs of BA (**a**), volume (**b**), and AGB (**c**) with scatterplot displaying correlation between observed and predicted values for the RF models.

## 4. Discussion

Models predicting BA, volume, and AGB in this study ranged in accuracy from $R^2 = 0.36$ to 0.53. In the RF models, the $R^2$ values were higher than those produced by the linear regression modeling approach, except for when predicting BA. Both RMSE and %RMSE were lower in linear regression models, except when predicting BA. Other researchers modeling forest metrics reported findings that RF modeling tends to have the smallest RMSE and the least amount of bias [36,65]. Because the predictive accuracy between the two modeling types was similar, the findings of this study suggest that the results of a modeling approach could depend more heavily on the variables used in modeling, as well as the forest type being modeled (mixed forest stands, homogeneous plantations). The models produced for volume had the greatest predictive accuracy ($R^2 = 0.53$, RMSE = 50.21 Mg ha$^{-1}$ %RMSE = 36.68), likely because the model used to calculate volume from field inventory included a height variable, while those for BA and AGB only included DBH and associated parameters. By calculating volume from field inventory using an equation that includes a height metric, lidar-derived height variables account for variation in heights when predictive modeling. Less accurate models developed for BA and AGB were also likely due to a number of other factors. For example, the difference in tree heights from when the field inventory was taken and when the lidar tree heights were measured could cause some variation; however, this is likely minimal as a subset of plots was used within two years of the lidar acquisition. The most likely explanation for the unexplained variation in BA and AGB models is due to a combination of the merchantable timber cruising process used to collect this field data and to the high heterogeneity of southeastern forests [34]. In other words, a lot of the foliage measured by lidar and multispectral imagery is not accounted for in the field inventory gathered by inventory cruisers. This is especially seen in the model building process for BA, where variables selected for each model were those whose function either distinguished layers of the canopy (i.e., density metrics trying to explain shrubs and non-inventoried trees beneath the tops of pines) or those that distinguished foliage type (texture bands). Height metrics were excluded from these models because they could not explain BA for all of the foliage within a plot, especially when the data that the models were built from were based only on merchantable timber instead of the entirety of the foliage constituting the plot BA. Furthermore, while there is a relationship between a tree's height and its BA, in many species, this relationship is known to be somewhat weak [28,38], and another reason why there are no height metrics in the model for BA. Similar findings were reported by other researchers working in heavily mixed forest stands, such as those in Canada, where BA was modeled with an $R^2$ of only 0.093 [28]. In that particular study, the researchers noted that it was difficult to account for most of the vegetation not measured by timber cruisers and recommended using another data source in addition to lidar.

From the models developed in this study, spectral, textural, and landcover variables were considered in addition to lidar to delineate any attributes of the forest not accounted for in the field inventory. As mentioned in the results section, the model for BA included a texture variable (var_3). Furthermore, the four and five variable models for volume and biomass included texture variables (var_2, hom_1). This suggests that in heavily mixed forests, spectral, textural, and landcover class variables have the potential to explain variation missed by lidar variables alone; however, the difference in model accuracy after removing imagery variables was not substantial, and lidar variables alone predicted BA, volume and AGB almost as well. This is not to say that lidar variables are sufficient in estimating lidar in Alabama mixed-species forests, as the accuracy of lidar-derived models were still poor in BA, volume, and AGB. While many papers suggest that lidar alone is sufficient for modeling, most of these studies are developed for study sites whose forests are primarily homogeneous such as those in the western United States [59,66,67]. In estimating BA, for example, a study in western Oregon achieved very high predictive accuracy using a model with two height variables and a canopy cover variable ($R^2 = 0.96$), but the study site was homogeneous in nature, dominated by old-growth coniferous forests [30]. Because of

the variation in allometric dimensions among tree species in mixed forest stands, alternative variables are necessary to accurately predict forest metrics.

## 5. Conclusions

RS data, including airborne and spaceborne lidar, as well as multispectral imagery from satellite-based platforms, are becoming a more available resource to foresters and ecologists. Programs such as the 3D Elevation program from the USGS, the National Agricultural Imagery Program from the USDA, and Sentinel-2 from the ESA provide free data that foresters can use to estimate the amount of BA, volume, and AGB on their stands and across large tracts of forested land. In this study, the potential of estimating BA, volume, and AGB from RS data was demonstrated by means of linear regression analysis and random forest modeling. Predictive accuracies of models were low ($R^2$ = 0.36–0.53) relative to those in some studies [27,59], and the results presented here suggest that the addition of imagery variables to lidar derived predictive models do not substantially help explain the variation in BA, volume, and AGB. While this study demonstrates the potential for a fast and efficient method of estimating BA, volume, and AGB using freely available data, further investigation of variables is needed to increase the variability of forest structure in mixed forest stands. Lastly, the main limitation of this study is that the field inventory used to build the models included merchantable timber and not the entire vegetation in one forest plot. Therefore, the potential to improve forest metric estimates may be improved if the entirety of the vegetation in a forest plot is measured.

Open-source RS data are increasingly available, and as demonstrated in this study, freely available data can be leveraged for estimating BA, volume, and AGB in a spatially explicit manner. Lastly, improving estimates of BA, volume, and AGB are necessary to produce accurate reference maps that can be used for the validation of forest measures derived from satellite lidar and imagery.

**Author Contributions:** Conceptualization, S.B. and L.L.N.; methodology, S.B. and L.L.N.; formal analysis, S.B.; investigation, S.B.; resources, S.B. and L.L.N.; writing—original draft preparation, S.B.; writing—review and editing, S.B., L.L.N., and J.G.; visualization, S.B.; supervision, L.L.N. All authors have read and agreed to the published version of the manuscript.

**Funding:** This research received no external funding.

**Data Availability Statement:** Forest inventory data is owned by the SDFEC and may be available upon request. All other data are publicaly available. 3DEP idar data can be found at https://prd-tnm.s3.amazonaws.com/LidarExplorer/index.html#/. Sentinel data can be found at https://earthexplorer.usgs.gov/.

**Acknowledgments:** We acknowledge and thank the SDFEC, a part of Auburn University, for collecting and providing the forest inventory data used in this study.

**Conflicts of Interest:** The authors declare no conflict of interest.

## Appendix A

**Table A1.** The best models produced for models of one to five variables for BA, volume, and AGB.

| Forest Metric | Model | %RMSE | RMSE | MAE | $R^2$ | Bias |
|---|---|---|---|---|---|---|
| $BA_1$ | BA = 2.15 + 0.0101cancovA | 37.44 | 6.865 | 5.453 | 0.16 | 1.333 |
| $BA_2$ | BA = 1.93 + 0.0164cancovA − 2.62dens_0_10 | 32.81 | 6.016 | 4.741 | 0.31 | 1.056 |
| $BA_3$ | BA = 1.94 + 0.0171cancovA − 1.95dens_0_10 − 0.910dens_10_15 | 31.91 | 5.850 | 4.584 | 0.34 | 1.005 |
| $BA_4$ | BA = 1.86 + 0.0181cancovA − 2.06dens_0_10 − 1.017dens_10_15 + 0.0240var_3 | 31.59 | 5.792 | 4.591 | 0.36 | 0.9676 |
| $BA_5$ | BA = 1.78 + 0.0173cancovA − 1.73dens_0_10 − 1.06dens_10_15 + 0.672dens_15_20 + 0.0245var_3 | 31.26 | 5.371 | 4.456 | 0.36 | 0.9528 |
| $Volume_1$ | Vol = 3.45 + 0.0797p50 | 39.92 | 55.75 | 42.30 | 0.32 | 11.85 |
| $Volume_2$ | Vol = 2.53 + 0.0892p50 + 0.0115cancovA | 36.35 | 50.77 | 37.72 | 0.45 | 8.893 |
| $Volume_3$ | Vol = 2.82 + 0.0719p50 + 0.0132cancovA − 0.957dens_10_15 | 35.97 | 50.24 | 37.13 | 0.45 | 8.698 |
| $Volume_4$ | Vol = 3.20 + 0.0721p50 + 0.0148cancovA − 1.11dens_10_15 − 0.594hom_1 | 35.32 | 49.33 | 37.30 | 0.45 | 8.299 |
| $Volume_5$ | Vol = 3.09 + 0.0769p50 + 0.0145cancovA − 1.03dens_10_15 − 0.590hom_1 + 0.213Elev_CV | 35.30 | 49.23 | 37.32 | 0.45 | 8.287 |
| $AGB_1$ | AGB = 3.63 + 0.00989cancovA | 40.01 | 32.20 | 24.88 | 0.15 | 6.368 |
| $AGB_2$ | AGB = 2.52 + 0.0122cancovA + 0.0568p50 | 32.65 | 26.27 | 19.88 | 0.38 | 4.434 |
| $AGB_3$ | AGB = 2.70 + 0.0132cancovA + 0.0464p50 − dens_10_15 | 32.46 | 26.12 | 19.65 | 0.38 | 4.387 |
| $AGB_4$ | AGB = 2.57 + 0.0141cancovA + 0.0484p50 − 0.626dens_10_15 + 0.0534var_2 | 31.95 | 25.72 | 19.59 | 0.39 | 4.160 |
| $AGB_5$ | AGB = 2.53 + 0.0609p50 + 0.0136cancovA + 0.924dens_15_20 + 0.0406var_2 − 0.432hom_1 | 31.26 | 25.20 | 19.35 | 0.41 | 4.064 |

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
