# Peer review of "Using Airborne Lidar, Multispectral Imagery, and Field Inventory Data to Estimate Basal Area, Volume, and Aboveground Biomass in Heterogeneous Mixed Species Forests: A Case Study in Southern Alabama"

_remotesensing, doi:10.3390/rs14112708_

Round 1

Reviewer 1 Report

Review of the manuscript: remotesensing-1719172

Using airborne lidar, multispectral imagery, and field inventory data to estimate basal area, volume, and aboveground biomass in heterogeneous mixed species forests: a case study in Southern Alabama (authors: Brown S., Narine L., Gilbert J.)

The manuscript presents a case study of using airborne Lidar and Sentinel 2 multi-spectral images to produce wall to wall maps of three forest metrics: Basal Area, Volume and Above Ground Biomass in mixed species stands. ­The main goal of the study were: 1) to investigate whether the performance of models based only on Lidar can be improved by adding multi-spectral imagery variables; ii) to select the variables useful for the specific modeling purposes. Using forest inventory data as response variables, two different approaches were tested to derive models of these metrics: multiple linear regression and random forests. For regression modeling, the dataset was split in 70% for training and 30% for testing.

General Comments

The manuscript is interesting and falls fully within the scope of the Journal. The topic is contextualized with a rich and updated literature review and the study objectives are well defined. The abstract is sufficiently informative, and keywords are appropriate. The method description is well structured, but equations are not clearly written: in particular, superscript and subscript elements are not correctly represented. Equation 1 should be reformulated and explained better (see specific comments). Also, I suggest using shorter abbreviations for the variables in Table 4, to improve the readability of the equations in the results. The method applied, as a whole, is appropriate, but bias assessment is missing. The results section is quite well structured, but I suggest presenting performances in terms of percentage errors also for Random Forest Model and adding some bias metrics. Discussion and conclusions are well organized and clear. Discussion can be enriched in case percentage errors and bias metrics are added to results. In conclusion, the main limitation of the study is well stressed: it is related to the characteristics of the data, whose survey protocol was more suitable for different purposes (timber cruising) than for the analysis presented.  

Specific comments:

Page 2, Line 49: the definition of AGB is not correct: it is not the Carbon content, but the weight of forest trees, including branches and twigs. Please, check the already cited paper of Jenkins et al. (2003)

Page 2, Line 56-58: I suggest avoiding to repeat “often”

Page 2, Line 90: the reference should be reported as a number in square brackets

Page 2, Line 95: see above

Page 7, Line 196: this equation should be rewritten. I suggest not to multiply for 10 each plot basal area (to obtain the value by acre), but to directly use a conversion factor from 0.04 hectare (plot) to 1 hectare. Moreover, another conversion factor (from feet squared to meters squared) should be added.

Page 7, Line 197: the constant 0.005454 it is not only a conversion factor from inches to feet squared, it also includes the Greek pi, it must be explained.

Page 7, Line 208-209: I suggest moving the explanation of “dbh” and “Ht” after the equation 1 and the equation 2, respectively.

Page 8, Line 241: the acronym CHM should be spelled out

Page 9, Line 256: the language to be cited is R. R-Studio is an integrated development environment for R

Page 9, Line 263: if leaps is an R package, it should be added

Page 10, Line 279: raster is a package of R, not R-Studio (is an integrated development environment for R). To cite R in publications use: R Core Team (2021). R: A language and environment for statistical computing. R Foundation for Statistical Computing, Vienna, Austria. URL https://www.R-project.org/.

Page 10, Line 287: ModelMap is an R package; it should be specified.

Page 10, Line 288: some explanations about mtry and ntree parameters could help the reader

Page 10, Line 301: "selected" is repeated

Page 11, Line 318: "max, and canopycover" should be written as in Table 4 "Max, and Canopycover"

Page 11, Line 324: I suggest adding the percentage RMSEs to this table

Page 12-13, Figure 3-4: for each metric, the maps obtained through the two different approaches  (linear regression and RF) are not comparable, because the palettes don't correspond to the same range of values

Page 14, Line 361-362: this is a good point. However, in Materials it has been specified that only a subset of inventory plots was selected," within two years of the 2017 Lidar and imagery data acquisition"

Author Response

Dear Reviewer 2,

Thank you for your response to this work.

I have made edits to the paper. 

Thank you for your contribution.

Reviewer 3 Report

This study is a try to model forest parameters as basal area, volume and above ground biomass, on mixed forest area in Southern Alabama, using lidar and optical data based on open source data.

The manuscript is not well written, and it needs work to be a completed scientific paper. The authors focus mostly on a statistical analysis , the results are not well presented, the discussion is not well structed, the method used is very common, the contribution and the innovation of this study is not well justified.

I consider their analysis as preliminary, since there are several issues to consider

  • Line 19-23 there is repeated information
  • Line 28-29 very general and already known fact
  • Keywords are replicated in the title
  • Line 36:The measures of BA used to managed…not BA used to managed
  • Line 39: you should refer other ecological studies in global level
  • Line 43: use and the latin name of Red-Cockaded woodpeckers
  • Line 46-46 unnecessary
  • Line 75-76, needs a reference
  • Line 95 no clear meaning
  • Line 123-125, however your study area was only 33.35km2
  • Line 119-125 the paragraph should be moved before study’s goal description
  • Line 128-140: You should describe the ecoregions or you should refer the main species of the area, the land cover, the temperature, the precipitation amount
  • Figure 1: the field plots points are hardly seen, what image is the background image
  • In general, be careful of the superscripts
  • Line 173-182 you give general information, and you miss major information, such as what exact dates of Sentinel 2 images are used, Surface reflectance or TOA, what level, what collection etc. Why you do not use the other bands?
  • Line 186: so you use the mean value of pixels inside the boundaries of each plot (0.04ha)?, please clarify. Why you do not use a window at least of 3*3 pixels
  • Figure 2: the graph is too big
  • Line 196: Explain again what DBH stands for and refer its measure unit (m or cm). Is dbh the same with DBH?
  • In Equation forms you should use × instead of *, be careful of superscript and use × wherever is needed
  • Line 229-231 make reference to table 4
  • Line 238-248 : why you didn’t use only NDVI as spectral variable. SWIR band contribute both to forest parameter estimation and to land cover classification. What was the accuracy of your classification? Did SDEFEC forest inventory not provide information of land cover of each plot, or of tree species measured?
  • Line 262-279. How many models were developed? How many models were removed because of VIF values? What variables present collinearity.
  • Line 281-289. You should describe random forest algorithm more and explain the hyperparameters of the algorithm.
  • Line 292-304: You should provide all models results (accuracy metrics, p value, VIF values, etc ) on a table attached at appendix.
  • Line 301: delete “was selected”
  • Line 327-335 Present all best models (both linear and RF) on one table.
  • Line 331-335 Did you compute variable importance? Which metric did you use? You should present a plot of variable importance.
  • Figure 3-4-5 you should present R2 and RMSE values
  • Discussion is one too long paragraph, very hard to follow. You should change the structure of discussion section and gather the information of result section
  • Line 415-418 there was no research on it.
  • Line 415-418. These are hypothesis, and not conclusions.
  • The results and the conclusions do not fully answer to the objectives of the study
  • The references are not all correct. For example, check 7-8-9-13

Reviewer 4 Report

General Comments

This study aims to produce more accurate wall-to-wall reference maps in mixed forest stands by introducing variables from multispectral imagery into lidar models. A total of 523 plots were used and 52 variables derived from Sentinel-2 multispectral imagery and 3D elevation program (3DEP) lidar were included in modeling to produce wall-to-wall estimates of BA, volume, and AGB in mixed species forests in Southern Alabama. The results showed that only the BA model included a variable for multispectral imagery, while volume and AGB models with only three lidar variables were sufficient. The sample size is large enough, the method is almost correct, and the results are relatively reliable. But I still hope the authors to revise the manuscript according the following comments.

  1. About linear and nonlinear regression models. Generally, nonlinear regression model is better than linear one. And recent studies also showed nonlinear model performed better than linear model. Thus, it is suggested develop nonlinear models based on 3- or 4-variables linear models.
  2. About the heteroscedasticity. In the section of regression modeling, the authors had paid attention to the multicollinearity, but not to heteroscedasticity. It is well known that the data of BA, volume and AGB exhibit heteroscedasticity. To deal with this problem, logarithmic regression or weighted regression must be used. The scatterplots in Figure 3 and 4 showed that the models are biased on the two sides, that is, over-estimated for small values and under-estimated for large values.
  3. About model validation. In this study, the authors applied the commonly used traditional method, 70% samples for training, and 30% for testing or validation. In fact, we can have so many alternatives of 70% samples, and the selection of the authors was only one of them. Of cause, not the best one. This is why the best model for training data did not perform well for testing data. In theory, the best one is based on all samples. For the validation of the model based on all samples, 5-fold or 10-fold method can be used.
  4. About model evaluation. In this study, the authors used two key statistics (R^2 and RMSE) for model assessment. It is not enough. For model evaluation, especially for comparison, the indices reflecting relative error, such as RMSE%, TRE, MPE, and MPSE, are more important. In addition, the residual error diagram is also very helpful.
  5. About the contribution of variables. In the three models in table 5, only the BA model included a variable from multispectral imagery. If we remove it, how much the R^2 would be reduced. Similarly, in the volume and AGB models, what are the contribution percentages of the three variables? They may be 80%, 15%, and 5%. This information is valuable for giving conclusion and suggestion.

The following paper can be referenced:

http://dx.doi.org/10.1016/j.rse.2014.10.004

Special Comments

  1. Abstract: The two modeling approaches, multiple linear regression and Random Forest (RF), should be stated, and regression model being better than the RF model also need to be presented.
  2. Line 114 and 117: The number 1 and 2 need to be replaced with a and b.
  3. Line 155-156: For clear understanding, it is suggested add “(4.6 inches)”, “(5.0 inches)” and “(3.0 inches)” after 11.68cm, 12.7cm and 7.62cm, respectively.
  4. Line 196 and 202: The unit of DBH in eq.1 seems to be inch, and the unit of dbh in eq.2 is cm, which need to be pointed out. DBH and dbh should be same, and the power exponent 2 must be superscript. The “i” should be subscript, and located after dbh in eq.1 and after dbh and Ht respectively in eq.2. In addition, there are many errors of expression such as m2ha-1 and m3ha-1 in the text.
  5. Table 2: The parameter β has only 4 significant digits, then parameter α also need to keep 4 significant digits, as -0.4432 and 0.8235. The rule keeping same decimal digits is not reasonable. If possible, it is better to keep 5 significant digits for α and β, consistent with the parameters in Table 3.
  6. Line 289-290: Here, it is suggested add a subsection “2.4.3 Model evaluation”.
  7. Table 5: All values in this table have 2 decimal digits, 2-5 significant digits, which might reveal the authors have no idea about significant digits. For the parameter estimates in the three models, except the intercept parameters, the slope parameters should keep same significant digits, at least 3 or 4 digits. In addition, in BA model, two variables are positive related, and other two negative; but in volume and AGB models, all three variables are positive related, no one variable negative. This is not normal. On my experience, the most negatively related variable should be included in the models.

Round 2

Reviewer 3 Report

The authors have not given satisfactory and clear answers, I hesitate to accept this paper as it is. However, thei work has been improved. I insist that the authors should know what satellite data they have used ( TOA or BOA), provide all models results on a table attached at appendix . How did they extract the spectral values? (the mean value of pixels inside the boundaries of each plot). Why they did not consider  more pixels (window3x3). What was the distance between plots? What was the classification accuracy?Bibliographic references should also be there about other ecological studies. All plots need to be improved. For example present only 10 of the most important variables. Figure 2 would be better on black and white (using different schemes for input-output- process).  And minor note: the point of keywords is to supplement the title information and not to replicate the information

Reviewer 4 Report

The authors have taken my comments into consideration for improving the manuscript. I have read the response to comments and reviewed the revised manuscript. I found the manuscript has been sufficiently improved, and the models were more reliable, which can be showed in Figure 4 and 5. On my opinion, the revised manuscript can be accepted for publication.
